# Effect of Different Raw Material Property for the Fabrication on Al/CNT Nanocomposite Using a Ball Mill with a Discrete Element Method (DEM) Simulation

**DOI:** 10.3390/ma12203291

**Published:** 2019-10-10

**Authors:** Battsetseg Jargalsaikhan, Amgalan Bor, Jehyun Lee, Heekyu Choi

**Affiliations:** 1Engineering Research Center (ERC) for Integrated Mechatronics Materials and Components, Changwon National University, Changwon, Gyoungnam 641-773, Korea; battsetseg12@yahoo.com (B.J.); amgalanbor@gmail.com (A.B.); ljh@changwon.ac.kr (J.L.); 2Graduate School of Material Science Engineering, Changwon National University, Changwon, Gyoungnam 641-773, Korea; 3Department of Mechanics Convergence, College of Engineering, Changwon National University, Changwon, Gyoungnam 641-773, Korea

**Keywords:** aluminum-based matrix composites, traditional ball mill, DEM simulation, mechanical alloying

## Abstract

Carbon nanotubes (CNTs) have received interest as an attractive reinforcing agent metal matrix composites regarded as an increase to mechanical properties of the final product. Aluminum/carbon nanotubes (Al/CNTs) nanocomposites were observed with different raw material at the optimized experimental condition. In this study, Al-based CNTs composites were three different samples, including un-milled Al, un-milled Al with CNTs, and milled Al with CNTs nanocomposites in the presence of additional CNTs with various experimental conditions while using a traditional ball mill (TBM). The particle morphology and CNT dispersions of milled composites were respectively analysed by scanning electron microscopy (SEM) and field emission scanning electron microscopy (FESEM), and the mechanical properties of the fabricated composites were tested. In each sample, CNTs were well dispersed on the surface of Al powder at different experimental conditions for milling in a TBM. The Al/CNTs nanocomposites were processed by compacting, sintering and rolling process. The Vickers hardness was used to characterize the mechanical properties. The hardness of Al/CNTs nanocomposites that were fabricated with milled Al with CNT was higher than the reached to in the nanocomposites prepared with the use of un-milled Al with CNT nanocomposites. Therefore, the discrete element method (DEM) simulation was used to complete quantitative analysis. The flow pattern, impact force, and energy at various experimental conditions are considered. The results of the simulations are compared with experimental data.

## 1. Introduction

Iijima discovered the carbon nanotubes (CNTs) in 1991 [1], and are they widely used for the fabrication of nanocomposite materials with metallic particles to improve the chemical, physical, and mechanical properties of the final products [2,3]. Choi et al. [4] studied the effect of structural and morphology of CNTs on the mechanical alloying characteristics of CNT-aluminum (Al) nanocomposite. They have studied Al powders with single-, double-, and multi-walled CNTs nanocomposites being prepared by ball mill. It can be found multi-walled CNTs (MWCNT) as reinforcements had best structural quality when compared to other types of the CNTs for nanocomposites. The composite of primary importance for studying the CNT dispersion under the different experimental condition and fabricating the metal-based CNT nanocomposites was owed to the excellent mechanical properties by the ball milling process [2,5,6]. 

When considering that aluminum has a relatively low density, low strength, and other mechanical properties, among many candidate matrix materials with lightweight and high strength composites. In the Al-matrix system, more difficulty increases due to the chemical reaction between Al and CNTs forming a compound Al_4_C_3_, practically at the processing temperature above approximately 800K [2,7,8]. Many researchers have studied carbon nanotube reinforced metal matrix composites; the addition of CNTs to the metal enhances the mechanical and physical properties by the mechanical alloying process. Wang et al. investigated interactions between CNTs and Al powder for the preparation of carbon/metal composite during mechanical alloying while using ball milling, and these have improved the hardness properties of the composite [9]. Bor et al. studied different raw material physical properties while using Cu/CNTs nanocomposites that were prepared using a planetary ball mill and investigated the effect of structural and morphology on the fabricated Cu/CNTs nanocomposites [3]. The common uses of CNT-based nanocomposites include modern electronic devices, solar cells, automobile, aerospace, and construction industries, owing to its lighter weight and easy deformability [10,11].

The discrete element method (DEM) simulation of ball mills is very useful as a tool for analysing the impact of energy and force [12]. The discrete element method considers particles individually and examining their motion and the force of contact between particles or between particles and mill walls are modeled, according to Newton’s laws of motion. The correlation between the impact energy and the grinding rate constant was estimated by DEM simulation [13,14].

In this study, we fabricate Al with CNTs nanocomposites with better the mechanical properties while using three different raw materials (un-milled Al without CNT, un-milled Al with CNTs, specially milled Al with CNT) by a traditional ball mill (TBM) while using a DEM simulation. In the article, the DEM is used to study the motion of ball charge in ball mills and powders when it is difficult to estimate detailed results that are impact force and energy from actual experiments. 

## 2. Experimental

Pure Al powders (purity: 99.8 %) with an average particle size of 44 μm by Alfa Aesar and the pristine MWCNTs (the Carbon Nano-material Technology Co. Ltd., Korea) with the diameter of 20 nm and length of 5μm were used in this study. Figure 1 shows the morphologies of the raw materials. 

We fabricated three different samples, including un-milled Al without CNTs powder (un-milled Al), un-milled Al with CNTs nanocomposites (un-milled Al with CNTs), and milled Al with CNTs nanocomposites (milled Al with CNTs) while using a traditional ball mill with various experimental conditions. Furthermore, un-milled Al means that the pure aluminum powder and milled Al means that pure Al powder was milled using a traditional ball mill during 30 min. at a rotation speed of 50 rpm. The pure Al and Al/CNT nanocomposites were already studied in a wide field. In this study, the common point is used to milled Al/CNT nanocomposite was compared to the un-milled Al and un-milled Al/CNTs nanocomposite. Milled Al powder is assumed that the eventual fractured crystal structure of the Al particles and dislocations on the Al particles occurs; the following CNT implies improved contact between the Al particles during the ball milling process. Mechanical alloying was carried out for the prepared three different samples by a traditional ball mill at the rotation speed of 100 and 200 rpm for milling time with 30 and 45 min. Table 1 summarizes the experimental conditions.

The fabricated composites were examined while using scanning electron microscopy (SEM) (JSM-6510, JEOL, Japan) for powder morphology and field emission scanning electron microscopy (FESEM) (CZ/MIRAI LMH, Tescan, Czech) for the dispersion of CNTs on the surface of Al powder. After milling, the fabricated composites are pressed in dies under high pressure to form them into the shape of 15 mm of height and 5 mm of depth while using 1.5 tons compacting machine made by Haji Engineering, Korea at room temperature. Subsequently, the conventional sintering of Al/CNTs composites was performed while using a vacuum tube machine, Haji Engineering, Korea. The sintering experimental conditions are shown in Table 2. Sintered Al/CNTs nanocomposites were rolled at room temperature by rolling machine, hardness has been observed. 

Table 3 gives the DEM simulation in the traditional ball mill was analyzed in the simulation condition.

The DEM simulation used in this work changed experimental condition is the milling rotation speed. The other parameters of simulation conditions are fixed. The ball mill is modeled by a discrete element method using the software package in the Samadii^TM^/DEM. The simulation time of three seconds represents the steady-state of the simulation, which is the reason the simulation experiment is short. It does not make sense to simulate long. 

## 3. Results and Discussion

Figure 2 shows the SEM photographs of un-milled Al without CNT powders was milled at the rotation speed of 100 rpm with 30 min. and 45 min. (Figure 2a,b) and rotation speed of 200 rpm with 30 min. and 45 min. (Figure 2c,d) by ball diameter of 5 mm. At first, we fabricated un-milled Al without CNT dramatically that did not change in morphology at 100 rpm and 200 rpm at different grinding times. The particle size was slightly increased in this experimental condition. It is for this reason that the low energy ball mill gave low energy impact to the powder. Choi et al. [15] discussed particle size does not change effectively for low rotation speed. In addition, a low rotation speed has less effect on powder crystallization due to collisions between particles in the ground powder at low rotation speed having less impact energy [16]. Figure 3 shows the change of particle morphology of un-milled Al with CNT composite at the rotation speed of 100 rpm and 200 rpm for different milling times of 30 min. and 45 min. while using a ball size of 5 mm. The particle morphology of un-milled Al with CNT nanocomposites was effectively not changed and slightly increased particle size. The SEM photographs milled Al with CNTs nanocomposites was milled experimental conditions with the rotation speed of 100 rpm and 200 rpm, ball diameter of 5 mm, and milling times of 30 min. and 45 min., as shown in Figure 4. Milled Al with CNTs nanocomposites did not change the morphology of particle and increases a particle size. The mechanical alloying process has two kinds of the process, when considering one is the cold-working of the powders that lead to a reduction in ductility and fracturing of the particles, the other is cold-welding of particles, which trends to increase the particle size [17]. 

We compared CNTs dispersion of un-milled Al with CNTs and milled Al with CNTs nanocomposites by FESEM results under the same experimental conditions. 

The FESEM micrographs in Figure 5 show the microstructures that were obtained when the un-milled Al with CNTs nanocomposites is processed with different rotation speeds are 100 rpm and 200 rpm and the milling times are 30 min. and 45 min. As the micrographs show, CNTs spread over the surface of Al powder during (Figure 5a) 30 min. at a rotation speed of (Figure 5a,b) 100 rpm and adhered and spread over the surface after (Figure 5b) 45 min. In the case of (Figure 5c,d) 200 rpm, the CNT coating that was obtained after (Figure 5c) 30 min. of milling showed to be well dispersed to that obtained after (Figure 5d) 45 min. 

Figure 6 shows the FESEM micrographs of CNTs dispersed in the surface of the milled Al. It can be seen that CNTs are adhered and well dispersed on the surface of milled Al with the rotation speed of 100 rpm at different milling times are (Figure 6a) 30 min. and (Figure 6b) 45 min. After milling with the rotation speed of (in Figure 6c,d) 200 rpm, the nanotubes are strongly attached and uniformly embedded in the Al surface or underlying CNTs under the mechanical force of milling balls. 

When compared with un-milled Al with CNTs, milled Al with CNTs nanocomposites has a significant amount of absorbed CNTs (the increasing area of the wrinkled imprint on the Al surface) and CNTs were better dispersed on the surface at long milling time. The collisions between powders at low rotation speed have less impact energy on the grinding media [2]. The essential factor in based on a metal matrix of CNTs particle coating is subjecting the sample to low-energy grinding ball impact force over a longer time, rather than higher force over a short time. That is the influence of the contact number during the milling process is greater than that of the impact power of the milling ball. Bor et al. [18] studied, by TBM, the best CNTs dispersions were for Cu/CNTs nanocomposites at a longer milling time at low rotation speed. 

We prepared from Al powder and Al-CNTs powder mixtures that measured the hardness of their samples by the Vickers hardness test. The hardness of samples, including un-milled Al, un-milled Al with CNTs, and milled Al with CNTs nanocomposites, are compared as experimental conditions of the rotation speed of 100 rpm and 200 rpm or milling times are 30 min. and 45 min. while using a traditional ball mill. 

The milled Al with CNTs nanocomposites is exactly increased to compare to others. Figure 7 shows the Vickers hardness results. The milled Al with CNTs nanocomposites hardness continues to increase with the rotation speed of 100 rpm and 200 rpm that hardness substantially higher than those of the other samples processed for the same time, reflecting the uniform dispersion of CNTs in the Al matrix. It is now needed to consider the reason for an improved mechanical property to the milled Al using for composite. Choi et al. [4] studied the composite samples of MWCNT-Al powder mixture, and their hardness increased with longer mechanical alloying time and showed hardness higher than that compared to other samples. The hardness test is an easy and simple method to measure the mechanical properties of materials. In the future study, we will study electrical, thermal conductivity, and high tensile strength.

Figure 8 shows a comparison between the results of observing the actual ball motion, which takes a camera and the results of analyzing the ball movement by simulation according to different rotation speeds are 100 rpm and 200 rpm while using the zirconia ball. The actual and simulated results show a similar trend between the experimental and simulated movements. 

We undertook quantitative analysis from DEM simulation, the impact force, and cumulative impact energy of the balls in a traditional ball mill with a different rotation speed could be obtained. DEM solves Newton’s equations of motion to resolve particle motion and uses a contact law to resolve inter-particle contact force [19]. Figure 9 and Figure 10 show two different rotation speeds of milling simulated. Note that all energy between balls and wall of the pot, among balls themselves, are considered. The results suggest that the material receives more strong impact from the balls for 200 rpm. In the simulation experiment, the sample does not get in the pot is regardless of the experimental results. The actual and simulated experiment results are potentially estimated, not matches.

## 4. Conclusions

In this study, we successfully fabricated three different samples of Al with CNT nanocomposites by observing the effect of various conditions on the physical properties. 

SEM results show that the particle morphology of Al powder was slightly increased with increasing rotation speed and milling time. The particle morphology was changed from an irregular type to kind of spherical type at 200 rpm in a milled Al with CNTs nanocomposites.

The FESEM results of fabricated nanocomposites show that CNTs were homogeneously dispersed over on the surface of the Al powder as well as successfully dispersed on the surface of Al powder milled Al with CNTs than un-milled Al with CNTs. 

The hardness of fabricated samples found that the hardness of the milled Al with CNTs was dramatically improved when compared to un-milled Al with CNTs under the same experimental conditions. 

The results of quantitative analysis from DEM simulation show a total force and energy that is not found in actual experiments.

## Figures and Tables

**Figure 1 materials-12-03291-f001:**
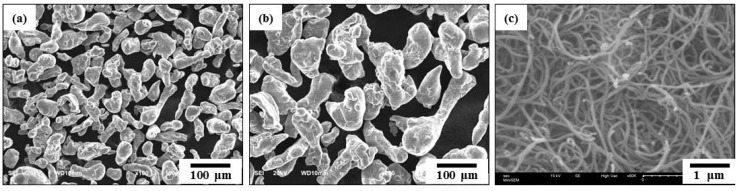
SEM micrographs (×200) of (**a**) the pure aluminum powder, (**b**) the milled aluminum powder, and (**c**) the MWCNTs starting materials used.

**Figure 2 materials-12-03291-f002:**
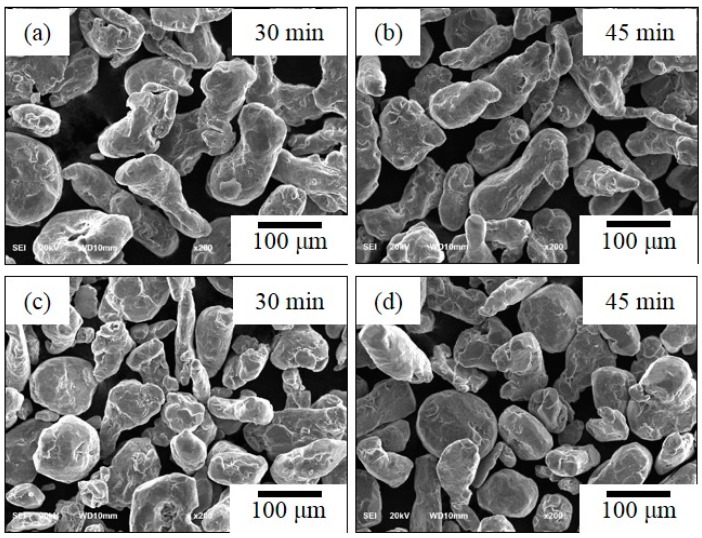
SEM photographs (×200), un-milled Al without CNT showing a rotation speed of (**a**,**b**) 100 rpm and (**c**,**d**) 200 rpm at different milling time of 30 min and 45 min.

**Figure 3 materials-12-03291-f003:**
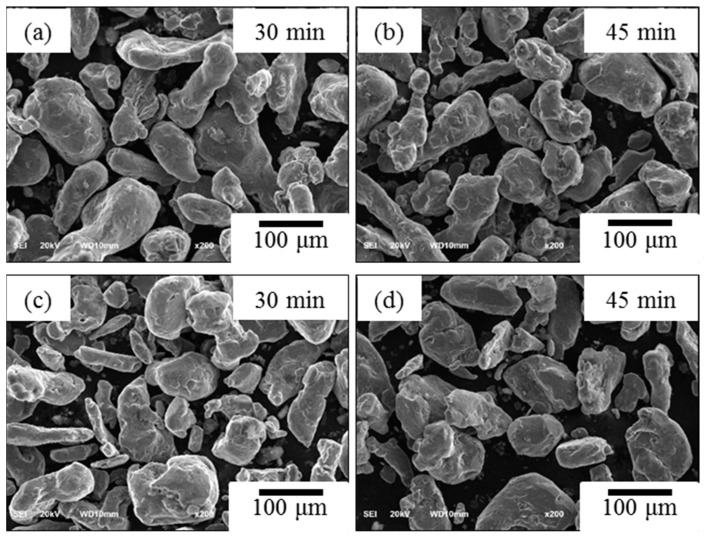
SEM photographs (×200), un-milled Al with CNT showing a rotation speed of (**a**,**b**) 100 rpm and (**c**,**d**) 200 rpm at different milling time of 30 min and 45 min.

**Figure 4 materials-12-03291-f004:**
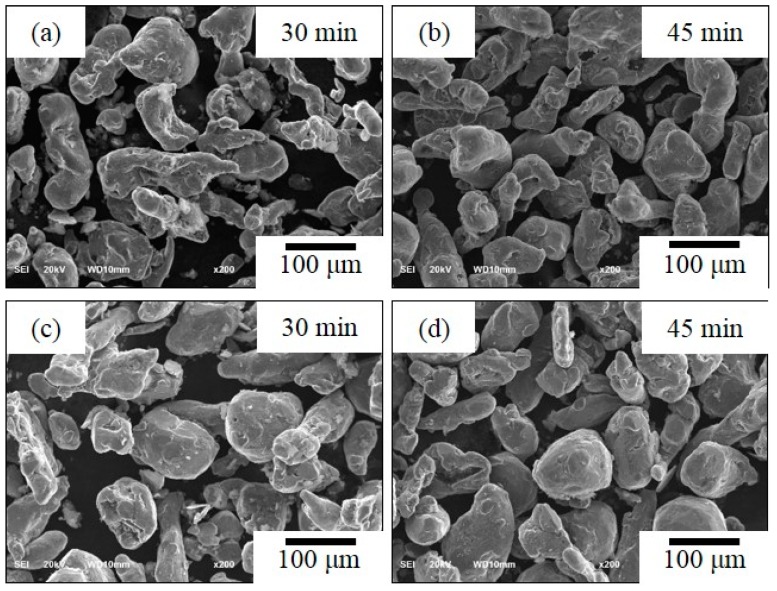
SEM photograph (×200), milled Al with CNT showing a rotation speed of (**a**,**b**) 100 rpm and (**c**,**d**) 200 rpm at different milling time of 30 min and 45 min.

**Figure 5 materials-12-03291-f005:**
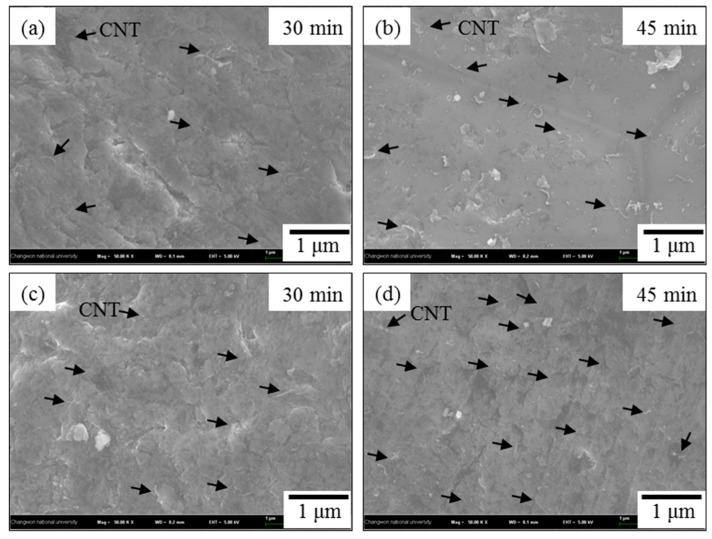
Field emission scanning electron microscopy micrographs (×50,000) of un-milled Al with CNT at (**a**,**b**) 100 rpm and (**c**,**d**) 200 rpm for various milling times with 30 min and 45 min. (black arrows indicate the CNTs).

**Figure 6 materials-12-03291-f006:**
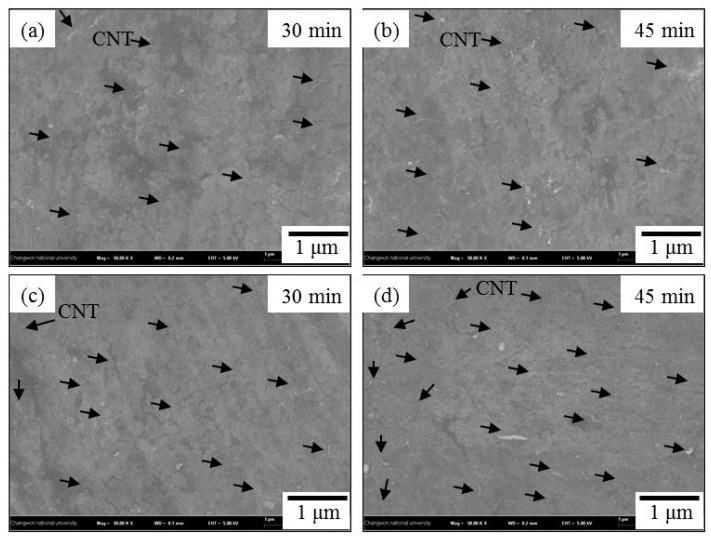
FESEM micrographs (×50,000) of milled Al with CNT at (**a**,**b**) 100 rpm and (**c**,**d**) 200 rpm for various milling times with 30 min and 45 min. (black arrows indicate the CNTs).

**Figure 7 materials-12-03291-f007:**
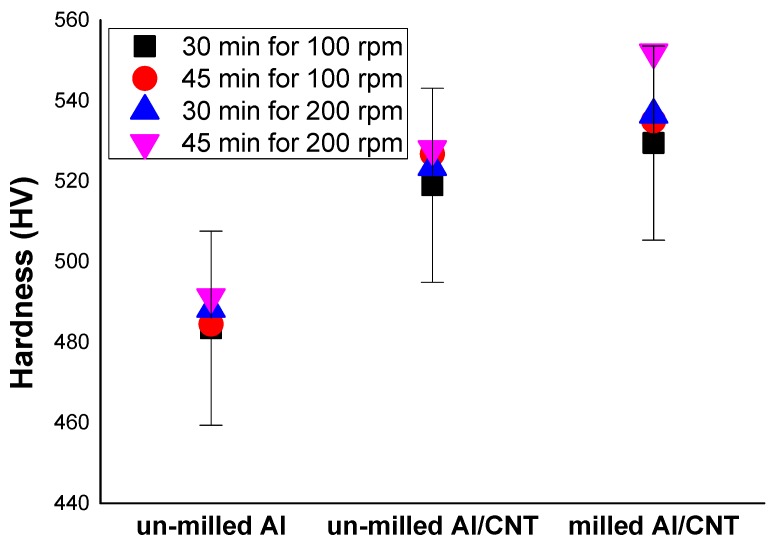
Results from Vickers hardness test. As a comparison, un-milled Al, un-milled Al with CNTs, and milled Al with CNTs under different milling time and rotation speed by a traditional ball mill.

**Figure 8 materials-12-03291-f008:**
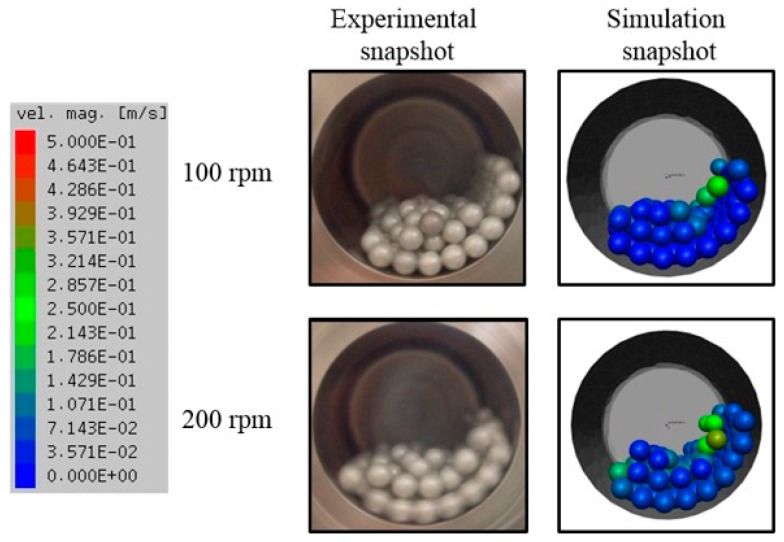
Actual photograph and simulation snapshot of the media motion results show from DEM simulation.

**Figure 9 materials-12-03291-f009:**
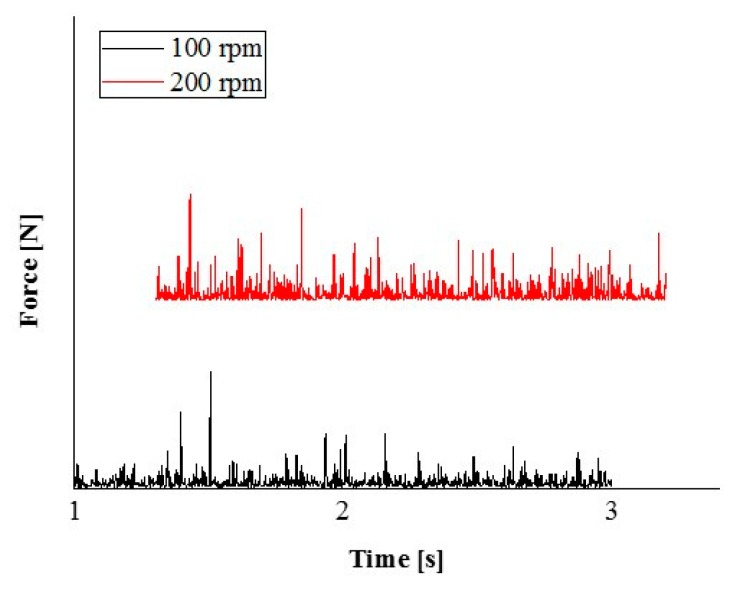
Impact force spectra of the balls in a traditional ball mill at different milling rotation speeds.

**Figure 10 materials-12-03291-f010:**
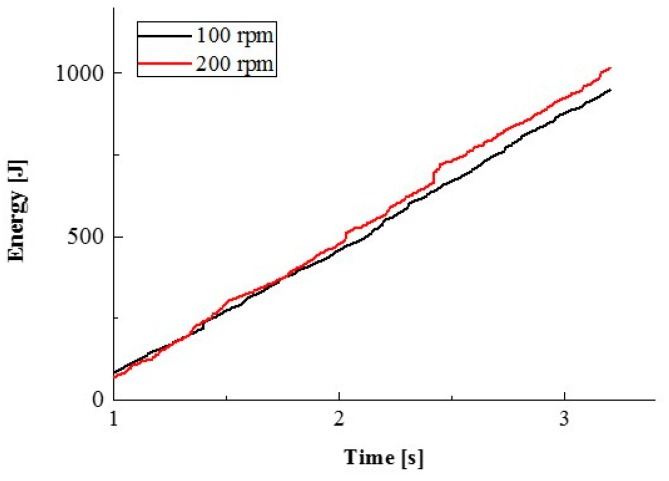
Cumulative impact energy distribution of the balls in a traditional ball mill at different milling rotation speeds.

**Table 1 materials-12-03291-t001:** Experimental conditions using three kinds of raw material for aluminum/carbon nanotubes (Al/CNTs) nanocomposite.

Experimental Conditions	
Ball milling equipment Rotation speed [rpm] Milling time [min] Ball diameter [mm] Ball powder ratio [-] Ball filling ratio [-] Amount of CNTs Material of media Temperature	planetary ball mill 100, 200 30, 45
	5 10:1 (fixed) 0.3 (fixed) 1 wt% zirconia room temperature

**Table 2 materials-12-03291-t002:** Experimental conditions of sintering.

Sintering Experimental Conditions	
Temperature [°C] Change time [min] Maintenance time [min]	500 150 240

**Table 3 materials-12-03291-t003:** Calculation condition for simulation.

Parameters		
Types of ball		Zirconia ball
Density [kg/m^3^]		6220
Young modulus [MPa]		200,000
Possion’s ratio		0.3
Dynamic friction coefficient [-]	ball-ball	0.5
	wall-ball	0.1
Rolling friction coefficient [-]		0.02
Thermal conductivity [W/(m K)]		3.5
Rotation speed [rpm]		100; 200
Ball diameter [mm]		5
Number of balls		113
Ball filling ratio [-]		0.3
Simulation time [sec]		3

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
