# Peer review of "Effect of Different Raw Material Property for the Fabrication on Al/CNT Nanocomposite Using a Ball Mill with a Discrete Element Method (DEM) Simulation"

_materials, 2019, doi:10.3390/ma12203291_

Round 1

Reviewer 1 Report

This work presents hardness study of ball milled carbon nanotube-Aluminum alloys. The ball milling process was investigate to explain changes in alloy properties over time and milling rotation speed. Results are interesting and suitable for this journal. However improvements are needed before it can considered for publication. My comments are:

1. Introduction: Sentences describing why this new study complements the field or add new information are missing in the Introduction. Some minor language check is needed, for instance, lines 49 and 50 "and these have been improved hardness" should read "and these have improved hardness".

2. Figure 7: are there error bars for the hardness results?

Figure 9: is this graph really providing a distribution curve? It should be labeled differently, e.g. cumulative energy over time.

3. Moreover, simulation time and experiments times are different. The former provide information for the first 3 minutes, while the latter goes up to 45 minutes. Any reason why simulations were cut short?

4. Overall, a better discussion providing the correlation between experimental results and simulations should be provided.

Author Response

Response to the Editor’s Concerns

Ref. No.: materials-600522

Title: Effect of different raw material property for the fabrication on Al/CNT nanocomposite using a ball mill with a discrete element method (DEM) simulation

Battsetseg Jargalsaikhan, Amgalan Bor, Jehyun Lee, Heekyu Choi

Dear Editor

We are grateful to the editor for the valuable comments regarding the accuracy of the data. Now the paper is further revised to include the corrections in accordance with the editor’s comments. Revisions are marked on the manuscript file in bold and yellow background.

Comments to authors by Editor:

Reviewer's comments:

This work presents hardness study of ball milled carbon nanotube-Aluminum alloys. The ball milling process was investigate to explain changes in alloy properties over time and milling rotation speed. Results are interesting and suitable for this journal. However improvements are needed before it can considered for publication. My comments are:

Introduction: Sentences describing why this new study complements the field or add new information are missing in the Introduction. Some minor language check is needed, for instance, lines 49 and 50 "and these have been improved hardness" should read "and these have improved hardness".

Response 1: We did correct in the introduction section

In this study to fabricate Al with CNTs nanocomposites with better mechanical properties using three different raw materials (un-milled Al without CNT, un-milled Al with CNTs, specially milled Al with CNT) by a traditional ball mill using a DEM simulation.

Figure 7: are there error bars for the hardness results?

Response 2: We indicated error bars on a graph of hardness in the results and discussion section.

Figure 9: is this graph really providing a distribution curve? It should be labeled differently, e.g. cumulative energy over time.

Response 3: We did correct in Figure 9.

Figure 9. Cumulative impact energy distribution of balls in the traditional ball mill at different milling rotation speeds.

Moreover, simulation time and experiments times are different. The former provide information for the first 3 minutes, while the latter goes up to 45 minutes. Any reason why simulations were cut short?

Response 4: The simulation time of 3 seconds represents the steady-state of the simulation, which reason is the simulation experiment is short. It does not make sense to simulate long.

Overall, a better discussion providing the correlation between experimental results and simulations should be provided.

Response 5: In the article, the DEM is used to study the motion of ball charge in ball mills and powders when it is difficult to estimate detailed results which are impact force and energy from actual experiments. By the simulation experiment, the sample does not get in the pot is regardless of the experimental results. Actual and simulated experiment results are potential estimated, not matches.

Thank you so much for your kind comments. 

Reviewer 2 Report

In the present state, this paper can not be appropriately read because of bad English. My related comments are in attached file.

Author Response

Response to the Editor’s Concerns

Ref. No.: materials-600522

Title: Effect of different raw material property for the fabrication on Al/CNT nanocomposite using a ball mill with a discrete element method (DEM) simulation

Battsetseg Jargalsaikhan, Amgalan Bor, Jehyun Lee, Heekyu Choi

Dear Editor

We are grateful to the editor for the valuable comments regarding the accuracy of the data. Now the paper is further revised to include the corrections in accordance with the editor’s comments. Revisions are marked on the manuscript file in bold and yellow background.

Comments to authors by Editor:

Reviewer's comments:

 Page 1

Al/CNTs nanocomposites were produced by powder metallurgy techniques on different property of raw material with an optimized condition

This sentence is unclear. CNT should be defined on the first use.

Response 1: CNTs have received interest as an attractive reinforcing agent metal matrix composites regarded as an increase to mechanical properties of the final product. Aluminum/ carbon nanotubes (Al/CNTs) nanocomposites were observed with different raw material at the optimized experimental condition.

We profitably fabricated 3 different samples including un-milled Al, un-milled Al with CNTs, and milled Al with CNTs nanocomposites in the presence of additional CNTs with various experimental conditions using a traditional ball mill (TBM).

Response 2: We corrected this fault in the abstract section.

In this study, Al-based CNTs composites were fabricated three different samples including un-milled Al, un-milled Al with CNTs, and milled Al with CNTs nanocomposites in the presence of additional CNTs with various experimental conditions using a traditional ball mill (TBM).

Particle morphology for a scanning electron microscopy (SEM) and CNT dispersions for a field emission scanning electron microscopy (FESEM) of milled composites were analysed and the mechanical properties of the fabricated composites were tested.

Response 3: We corrected this fault in the abstract section.

The particle morphology and CNT dispersions of milled composites were respectively analyzed by scanning electron microscopy (SEM) and field emission scanning electron microscopy (FESEM), and the mechanical properties of the fabricated composites were tested.

CNTs were well dispersed on the surface of Al powder each sample at various experimental conditions for milling in a TBM.

Response 4: We corrected this fault in the abstract section.

In each sample, CNTs were well dispersed on the surface of Al powder at different experimental conditions used for milling in TBM.

The Vickers hardness used to characterize the mechanical properties.

Response 5: We corrected this fault in the abstract section.

The Vickers hardness was used to characterize the mechanical properties.

The hardness of Al/CNTs nanocomposites was improved to milled Al with CNT than to un-milled Al with CNT nanocomposites.

Response 6: We corrected this fault in the abstract section.

The hardness of Al/CNTs nanocomposites fabricated with milled Al with CNT was higher than the reached in the nanocomposites prepared with the use of un-milled Al with CNT.

Therefore DEM simulation used to do quantitative analysis

What is a “DEM”?

Response 7: We corrected to indicate the description of DEM.

Therefore the discrete element method (DEM) simulation used to do quantitative analysis.

Flow pattern, impact force and energy on various experimental conditions are considered. Results of simulations are compared with experimental data.

Response 8: We corrected this fault in the abstract section.

The flow pattern, impact force and energy at various experimental conditions are considered. The results of the simulations are compared with experimental data.

The carbon nanotubes (CNTs) discovered by Iijima in 1991 [1], it have been widely used for fabrication of nanocomposite materials with metallic particles to improve the chemical, physical and the mechanical properties of the final products [2, 3].

Response 9: We corrected this fault in the abstract section.

The carbon nanotubes (CNTs) were discovered by Iijima in 1991 [1], and they are widely used for fabrication of nanocomposite materials with metallic particles to improve the chemical, physical and the mechanical properties of the final products [2, 3].

Choi et al. [4] studied effect of structural and morphology of CNTs on the mechanical alloying characteristics of CNT-Al powder mechanical properties of final products were studied Al powders with single-, double-, and multi-walled CNTs nanocomposites prepared by ball milling, resulting multi-walled CNTs as reinforcements had best structural quality compare to other types of the CNTs for nanocomposites.

This sentence is unclear.

Response 10: Choi et al. [4] studied the effect of structural and morphology of CNTs on the mechanical alloying characteristics of CNT-aluminum (Al). They have studied Al powders with single-, double-, and multi-walled CNTs nanocomposites prepared by ball mill, resulting in multi-walled CNTs as reinforcements had best structural quality compare to other types of the CNTs for nanocomposites.

The composite of primary importance to study the CNT dispersion under different experimental condition and to produce the CNT/Al composites with good mechanical properties by optimizing the ball milling condition [2, 5, 6].

This sentence is unclear.

Response 11: The composite of primary importance to study the CNT dispersion under the different experimental condition and to fabricate the CNT-based Al composites with better mechanical properties by the ball milling process.

Considering aluminium (Al) has a relatively low density, low strength, and other mechanical properties, among many candidate matrix materials for lightweight, high strength composites.

This sentence is unclear.

Response 12: Considering aluminum has a relatively low density, strength, and other mechanical properties, among many candidate matrix materials with lightweight, high strength composites.

Thank you so much for your kind comments.

Reviewer 3 Report

The article has some useful information, but the authors need to do some serious work so that the article matches the level of the journal. Key notes:

In general, Abstract was written satisfactorily, but I would like to see more focused results. In the Conclusion, the results are sufficiently fully formulated, but in the Abstract they should also be indicated.

In the Introduction there is no formulation of the purpose of the work. There is a review of a number of articles, but there is no generalization, justification for the need for research and their direction.

In the photo (Figure 1) I do not see any objects similar to "diameter of 20 nm and length of 5μm".

What is the difference between ground aluminum powder and milled aluminum powder? Why was this raw material chosen?

Why was it selected three different samples? In theory, there should still be milled Al without CNTs. Again, there are absolutely no explanations - goals, objectives, why is this raw material exactly?

"rotation speed of 100 and 200 rpm for milling time with 30, 45 min" - so what was rotation speed and milling time? Or were 4 samples made with various combinations of these parameters? If there are 3 kinds of raw material, then there should be at least 12 experiments. It is necessary to explain and clearly describe the conditions of the experiments. It is best to designate all samples (for example, A1, B2 or in any other way) and at the beginning give a table with a description of the experimental design.

it would also be correct to justify why precisely such speeds and why such processing time was chosen.

Honestly, I do not see any noticeable differences between the images in Figures 2 - 4. Perhaps there are some differences, but they are not obvious. It is necessary to give a more detailed description, possibly to zoom in and highlight important elements. These figures now provide extremely little information.

Figures 5-6. The image scale should be significantly increased. Now it is not entirely clear what the black arrows indicate. I also recommend using other colors (for example, red) to highlight important objects.

Figures 5-6. There are no scale bars and it is unclear what "1 micron" refers to.

Why is only hardness tested? Is this the only significant parameter?

Why do we need a numerical simulation? What results were obtained with its help? What is their significance for science and practice?

And where is the description of the conditions for the numerical simulation? What goals were set?

Has the simulation result been verified? How? In particular, the data of Figures 9 and 10 - how much do they correspond to experimental data?

Author Response

Response to the Editor’s Concerns

Ref. No.: materials-600522

Title: Effect of different raw material property for the fabrication on Al/CpNT nanocomposite using a ball mill with a discrete element method (DEM) simulation

Battsetseg Jargalsaikhan, Amgalan Bor, Jehyun Lee, Heekyu Choi

Dear Editor

We are grateful to the editor for the valuable comments regarding the accuracy of the data. Now the paper is further revised to include the corrections in accordance with the editor’s comments. Revisions are marked on the manuscript file in bold and yellow background.

Comments to authors by Editor:

Reviewer's comments:

In the photo (Figure 1) I do not see objects similar to “diameter of 20 nm and length of 5µm”.

Response 1: We corrected indicate to SEM photography of the MWCNTs in Figure 1 description section.

What is the difference between ground aluminum powder and milled aluminum powder? Why was this raw material chosen?

Response 2: We have corrected explanation about figure 1. That was the fault.

Why was it selected three different samples? In theory, there should still be milled Al without CNTs. Again, there are absolutely no explanations - goals, objectives, why is this raw material exactly

Response 3: We corrected these problems in the experiment section.

The pure Al and Al/CNT nanocomposite already studied in a wide field. In this study, the common point is used to milled Al/CNT nanocomposite was compared to un-milled Al and un-milled Al/CNTs nanocomposite. Milled Al powder is assumed that the eventual fractured crystal structure of the Al particles and occur dislocations on the Al particles; the following CNT implies improved contact between the Al particles during the ball milling process.

In this study to fabricate Al with CNTs nanocomposites with better mechanical properties using three different raw materials (un-milled Al without CNT, un-milled Al with CNTs, specially milled Al with CNT) by a traditional ball mill using a DEM simulation. In the article the DEM is used to study the motion of ball charge in ball mills and powders when it is difficult to estimate detailed results which are impact force and energy from actual experiments

"rotation speed of 100 and 200 rpm for milling time with 30, 45 min" - so what was rotation speed and milling time?Or were 4 samples made with various combinations of these parameters? If there are 3 kinds of raw material, then there should be at least 12 experiments. It is necessary to explain and clearly describe the conditions of the experiments. It is best to designate all samples (for example, A1, B2 or in any other way) and at the beginning give a table with a description of the experimental design.

Response 4: Figure 2 shows SEM photographs of un-milled Al without CNT powders was milled at the rotation speed of 100 rpm with 30 min and 45 min (figure 2a and 2b) and rotation speed of 200 rpm with 30 min and 45 min (figure 2c and 2d) by ball diameter of 5 mm.

it would also be correct to justify why precisely such speeds and why such processing time was chosen.

Response 5: We used to do many experiments to find optimal experimental conditions. Al is soft and ductile. If we carry an experiment for a long time and powerful rotation speed, resulting in Al powder occurs agglomeration. The main problem of Al-based composites research is to occur agglomeration. If we carry an experiment for an over short time and powerless rotation speed, resulting in Al/CNT nanocomposite not fabricated. In this study, the chosen milling time and rotation speed are optimal experimental data.

Honestly, I do not see any noticeable differences between the images in Figures 2 - 4. Perhaps there are some differences, but they are not obvious. It is necessary to give a more detailed description, possibly to zoom in and highlight important elements. These figures now provide extremely little information.

Response 6: Actually, we did not change the morphology of SEM images at the experimental condition in this study. Therefore, we found the difference in compared with un-milled Al with CNTs, milled Al with CNTs nanocomposites has a lot of amount of absorbed CNTs (the increasing area of the wrinkled imprint on the Al surface) and CNTs better dispersed on the surface at long milling time in the FESEM images.

Figures 5-6. The image scale should be significantly increased.Now it is not entirely clear what the black arrows indicate. I also recommend using other colors (for example, red) to highlight important objects.

Response 7: The black arrows indicate dispersed amounts of CNT. Thank you for your kind recommendation.

Figures 5-6. There are no scale bars and it is unclear what "1 micron" refers to.

Response 8: We corrected the scale bar on FESEM images in Figure 5 and 6.

Why is only hardness tested?Is this the only significant parameter?

Response 9:The hardness test is an easy and simple method to measure the mechanical properties of materials. In the future study, we will study electrical, thermal conductivity and high tensile strength.

Why do we need a numerical simulation? What results were obtained with its help? What is their significance for science and practice?

And where is the description of the conditions for the numerical simulation? What goals were set?

Response 10: As a tool for analyzing the impact of energy and force, the discrete element method (DEM) simulation of ball mills is very useful [12]. The discrete element method is considered to particles individually and examines their motion and the force of contact between particles or between particles and mill walls are modeled, according to Newton’s laws of motion.

In the article, the DEM is used to study the motion of ball charge in ball mills and powders when it is difficult to estimate detailed results from actual experiments.

Has the simulation result been verified? How? In particular, the data of Figures 9 and 10 - how much do they correspond to experimental data?

Response 11: The DEM is used to study the movement of balls in the ball mill to compare real ball movement and powders when it is difficult to estimate detailed results impact force and energies from actual experiments.

 Thank you so much for your kind comments.

Reviewer 4 Report

The article would be interesting to the readers.

I have some remarks about simulation study:

- Which CAE software has been used in this study? when a simulation analysis is made, it is important to know the type of the product, which is used. Different software products have different capabilities and accuracy of calculations

- The initial conditions of the simulation analysis was not described.

What is the application of the manufactured nanocomposites? This information could be included in the conclusions.

Author Response

Response to the Editor’s Concerns

Ref. No.: materials-600522

Title: Effect of different raw material property for the fabrication on Al/CNT nanocomposite using a ball mill with a discrete element method (DEM) simulation

Battsetseg Jargalsaikhan, Amgalan Bor, Jehyun Lee, Heekyu Choi

Dear Editor

We are grateful to the editor for the valuable comments regarding the accuracy of the data. Now the paper is further revised to include the corrections in accordance with the editor’s comments. Revisions are marked on the manuscript file in bold and yellow background.

Comments to authors by Editor:

Reviewer's comments:

Which CAE software has been used in this study? when a simulation analysis is made, it is important to know the type of the product, which is used. Different software products have different capabilities and accuracy of calculations

Response 1: We corrected in the experimental section.

The ball mill is modeled by a discrete element method using the software package in the SamadiiTM/DEM.

The initial conditions of the simulation analysis was not described.

Response 2: We corrected in the experimental section.

DEM simulation used in this work changed experimental condition is milling rotation speed. Other parameters of simulation conditions are fixed. The simulation time of 3 seconds represents the steady-state of the simulation, which reason is the simulation experiment is short. It does not make sense to simulate long.

What is the application of the manufactured nanocomposites? This information could be included in the conclusions.

Response 3: We indicated information about the application of the manufactured nanocomposites in the introduction section.

The common uses of CNT-based nanocomposites are including modern electronic devices, solar cells, automobile, aerospace, and construction industries owing to its lighter in weight and easy deformability [10, 11].

Thank you so much for your kind comments.

Reviewer 5 Report

Generally, this manuscript may be considered original. The structure of this paper needs to be improved. I would like to recommend this paper for publication in the journal “Materials“ after solving some issues:

line 15: "We profitably..." Avoid the first person (We, I etc.) unless it is an absolute requirement. This applies to the entire manuscript.

line 24: "used" should be "was used".

line 25: "used" should be "was used". However it is better to say: "The Vickers hardness tests have been carried out to characterize..."

Formally, the all acronyms used have to be explained at first use. See "TBM", "DEM" in abstract; MWCNTs in section Experimental, etc. This applies to the entire manuscript.

lines 34-38: Sentence is grammatically incomprehensible.

In this sentence: "The composite of primary importance to study the CNT dispersion under different experimental condition and to produce the CNT/Al composites with good mechanical properties by optimizing the ball milling condition [2, 5, 6]." there is a lack of verb!

lines 48-50: Authors says "Wang et al ...", however, in the references list there is no paper written by Wang et al.

line 50: "Amgalan Bor et al. studied ..." should be "Bor et al. [3] studied..."

line 53. Following the context of the introduction section it should be clearer to say "In this article the discrete element ..."

line 57: "are show" should be "are shown".

line 81: "... was carried out was analyzed ..." ???

Check grammar of the entire manuscript! The writing, the structuring of sentences and the use of English language are on an insufficient level. The descriptions and explanations are illegible and difficult to understand. Please note that I indicated in review form only the main grammar errors.

To numbering the tables use 1, 2, 3 ... not I, II, III.

In chapter Experimental: Authors says "DEM simulation in traditional ball mill was carried out was analyzed in the simulation condition are given in Table III." But there is no specific description of numerical procedures used in the numerical model. By the way, the description of discrete element  modeling should be included in separate section. There is no information that the Authors used a commercial program based on the discrete element method. So, I assume that the Authors wrote own procedures which must be frank explained. Otherwise, it is difficult for the reviewer to assess whether the procedures used were correct.

The general procedures of DEM are well known and they are not explained in this paper. However, the application of DEM to analyse specific problem requires a specific approach. How do the Authors ensure the stability of numerical algorithm? What were the boundary conditions, i.e. friction model, etc...?

What are the values of milling force in figure 10?

line 136: Paper [14] is authored to "Bor et al." not "Amgalan et al."

lines 136-137: "Amgalan et al. [14] studied by TBM, the best coatings were for Cu/CNTs nanocomposites at longer milling time at 50 rpm." What did you mean, in the context of this paragraph?

line 144: "show" should be replaced by "are shown"

line 157: "We did quantitative analysis from DEM simulation results can be obtained..." ?

lines 159-160: "Figure. 9 and 10 shows..." should be "Figures 9 and 10 show ..."

Author Response

Response to the Editor’s Concerns

Ref. No.: materials-600522

Title: Effect of different raw material property for the fabrication on Al/CNT nanocomposite using a ball mill with a discrete element method (DEM) simulation

Battsetseg Jargalsaikhan, Amgalan Bor, Jehyun Lee, Heekyu Choi

Dear Editor

We are grateful to the editor for the valuable comments regarding the accuracy of the data. Now the paper is further revised to include the corrections in accordance with the editor’s comments. Revisions are marked on the manuscript file in bold and yellow background.

Comments to authors by Editor:

Reviewer's comments:

Generally, this manuscript may be considered original. The structure of this paper needs to be improved. I would like to recommend this paper for publication in the journal “Materials“ after solving some issues:

"We profitably..." Avoid the first person (We, I etc.) unless it is an absolute requirement. This applies to the entire manuscript.

Response 1: We corrected this fault in the abstract section.

In this study, Al-based CNTs composites were fabricated three different samples including un-milled Al, un-milled Al with CNTs, and milled Al with CNTs nanocomposites in the presence of additional CNTs with various experimental conditions using a traditional ball mill (TBM).

"used" should be "was used". However it is better to say: "The Vickers hardness tests have been carried out to characterize..."

Response 2: The Vickers hardness tests have been carried out to characterize the mechanical properties.

Formally, the all acronyms used have to be explained at first use. See "TBM", " DEM" in abstract; MWCNTs in section Experimental, etc. This applies to the entire manuscript.

Response 3We corrected these faults.

lines 34-38: Sentence is grammatically incomprehensible.

Response 4: Choi et al. [4] studied the effect of structural and morphology of CNTs on the mechanical alloying characteristics of CNT-aluminum (Al) nanocomposite. powder mechanical properties of final products were They have studied Al powders with single-, double-, and multi-walled CNTs nanocomposites prepared by ball mill, resulting It can be found multi-walled CNTs as reinforcements had best structural quality compare to other types of the CNTs for nanocomposites.

In this sentence: "The composite of primary importance to study the CNT dispersion under different experimental condition and to produce the CNT/Al composites with good mechanical properties by optimizing the ball milling condition [2, 5, 6]. " there is a lack of verb!

Response 5: The composite of primary importance to study the CNT dispersion under the different experimental condition and to produce fabricate the CNT-based /Al composites with better mechanical properties by optimizing the ball milling process condition [2, 5, 6].

lines 48-50: Authors says "Wang et al ...", however in the references list there is no paper written by Wang et al.

Response 6: We corrected 9th reference in the reference list.

Wang, H. Choi, J. Myoung, and W. Lee. Mechanical alloying of multi-walled carbon nanotubes and aluminum powders for the preparation of carbon/metal composites. Carbon. 2009, 47, 3427-3433.

 7. line 50: "Amgalan Bor et al. studied ..." should be " "Bor et al. [3] studied..."

Response 7: We corrected this fault in the introduction section.

line 53. Following the context of the introduction section it should be clearer to say "In this article the discrete element ..."

Response 8: In the article, the DEM is used to study the motion of ball charge in ball mills and powders when it is difficult to estimate detailed results which are impact force and energy from actual experiments.

line 57: "are show" should be "are shown".

Response 9: The morphologies of the raw materials are shown in Figure 1.

line 81: "... was carried out was analyzed ..." ???

Response 10: DEM simulation in traditional ball mill was carried out was analyzed in the simulation condition are given in Table 3.

To numbering the tables use 1, 2, 3 ... not I, II, III.

Response 11: We corrected this fault in the description of tables.

In chapter Experimental: Authors says "DEM simulation in traditional ball mill was carried out was analyzed in the simulation condition are given in Table III." But there is no specific description of numerical procedures used in the numerical model. By the way, the description of discrete element modeling should be included in separate section. There is no information that the Authors used a commercial program based on the discrete element method. So, I assume that the Authors wrote own procedures which must be frank explained. Otherwise, it is difficult for the reviewer to assess whether the procedures used were correct.

Response 12: DEM simulation in traditional ball mill was carried out was analyzed in the simulation condition are given in Table 3.

DEM simulation used in this work changed experimental condition is milling rotation speed. Other parameters of simulation conditions are fixed. The ball mill is modeled by a discrete element method using the software package in the SamadiiTM/DEM. The simulation time of 3 seconds represents the steady-state of the simulation, which reason is the simulation experiment is short. It does not make sense to simulate long.

What are the values of milling force in figure 10?

Response 13: We changed the location of Figure 9 and 10 images.

    Figure 10 9. Milling Impact force spectra of the balls in a traditional ball mill at different milling rotation speeds.

line 136: Paper [14] is authored to "Bor et al." not " Amgalan et al."

Response 14: We corrected in the results and discussion section.

lines 136-137: "Amgalan et al. [14] studied by TBM, the best coatings were for Cu/CNTs nanocomposites at longer milling time at 50 rpm." What did you mean, in the context of this paragraph?

Response 15: Amgalan Bor et al. [18] studied by TBM, the best coatings CNTs dispersion were for Cu/CNTs nanocomposites at longer milling time at 50 rpm. the low rotation speed.

line 144: "show" should be replaced by "are shown"

Response 16: Vickers hardness results are shown in Figure 7.

line 157: "We did quantitative analysis from DEM simulation results can be obtained..." ?

Response 17: We did quantitative analysis from DEM simulation, results it can be obtained the energy and impact force and cumulative impact energy of the balls at various rotation speeds by the in a traditional ball mill with different rotation speed.

lines 159-160: "Figure. 9 and 10 shows..." should be "Figures 9 and 10 show ..."

Response 18: We corrected this fault.

Thank you so much for your kind comments.

Round 2

Reviewer 1 Report

The work presenting the hardness study of ball milled multiwall carbon nanotube (MWNT)-Aluminum alloys has been satisfactorily revised according to the comments from the reviewer. In this work, the ball milling process was investigate to explain changes in alloy properties over time and milling rotation speed; simulation results help explain the optimum ball-milling conditions required to making composites with increased hardness. Results are interesting and suitable for this journal. It is recommended for publication with only minor spell/language revisions.

Author Response

Hello, Dear Editor

Thank you so much for your grateful editing process and kind comments.

Sincerely, 

Battsetseg Jargalsaikhan, Amgalan Bor, Jehyun Lee, Heekyu Choi

Reviewer 2 Report

After the revision, the paper is significantly improved and, in my opinion, it can be published in the present state.

Author Response

(The authors gave the same response as above.)

Reviewer 3 Report

The authors did a great job to improve the quality of the article. In its present form, the article is generally suitable for publication. There remains only one significant point that the authors need to solve: the connection between the first (experimental) part and the second part (modeling) remains very weak and unobvious. This problem can be eliminated by describing the relationship between process characteristics (modeled, for example, between Cumulative impact energy - Figure 10 and the grain structure). Without such a clear and understandable connection, the article splits into two parts, the relationship between which is not obvious.

Author Response

Hello, Dear Editor

Currently, the simulation commercial model used has a little difficulty in revealing the relationship between the simulation and the experiment described by the author. In the future, it will be a more in-depth simulation study.

Thank you so much for your grateful editing process and kind comments.

Sincerely, 

Battsetseg Jargalsaikhan, Amgalan Bor, Jehyun Lee, Heekyu Choi

Round 3

Reviewer 3 Report

After additional corrections made by the authors, the article can be recommended for publication, while the reviewer recommends that the authors in subsequent works pay more attention to the connection between modeling and experiments. Modeling does not make sense if it is not connected with reality!